# Adaptive Hybrid Surgery Experiences in Benign Skull Base Tumors

**DOI:** 10.3390/brainsci12101326

**Published:** 2022-09-30

**Authors:** Jenny Christine Kienzler, Javier Fandino

**Affiliations:** 1Department of Neurosurgery, Kantonsspital Aarau, Tellstrasse 25, 5001 Aarau, Switzerland; 2Department of Neurosurgery, Hirslanden Medical Center, 5001 Aarau, Switzerland; 3Department of Neurosurgery, Hirslanden Medical Center, 8032 Zurich, Switzerland

**Keywords:** meningioma, skull base tumor, adaptive hybrid surgery, stereotactic radiosurgery, intraoperative volumetry, surgical resection

## Abstract

Background: The treatment of benign skull base tumors remains challenging. These tumors are often located in close relationship to critical structures. Therefore, radical resection of these tumors can be associated with high morbidity. Multimodal treatment concepts, including controlled partial tumor resection followed by radiosurgery, should be considered. Methods: Adaptive hybrid surgery analysis (AHSA) is an intraoperative tool that has been introduced for the automatic assessment of tumor properties, and virtual real-time radiosurgical treatment simulation and continuous feasibility analysis of adjuvant radiosurgery. The AHSA method (Brainlab^®^, Munich, Germany) was applied to five patients who underwent partial resection of a benign skull base tumor. Tumor volumetry was obtained on pre- and postoperative MR scans. Organs at risk were, preoperative, automatically delineated with atlas mapping software (Elements^®^ Segmentation Cranial), and adaptations were made if necessary. Results: Five patients with benign skull base lesions underwent planned partial tumor resection in a multimodal therapeutic surgery followed by radiosurgery. The preoperative tumor volumes ranged between 8.52 and 25.2 cm^3^. The intraoperative residual tumor volume measured with the AHSA^®^ software ranged between 2.13–12.17 cm^3^ (25–52% of the preoperative tumor volume). The intraoperative automatic AHSA plans of the remaining tumor volume suggested, in all five patients, that safe hypofractionated radiation was feasible. Patients were followed for 69.6 ± 1.04 months, and no complications occurred after the patients were treated with radiation. Conclusions: Intraoperative SRS planning based on volumetric assessments during resection of skull base tumors using AHSA^®^ is feasible and safe. The AHSA method allows the neurosurgeon to continuously evaluate the feasibility of adjuvant radiosurgery while planning and performing a surgical resection. This method supports the treatment strategy of a complementary approach during surgical resection of complex skull base tumors and might contribute to preventing surgical and radiosurgical complications.

## 1. Introduction

Skull base tumors, although mostly benign, are often in close relationship to critical structures such as cranial nerves or the brainstem; thus, the treatment of these tumors has always been challenging for neurosurgeons. Due to their slow growth and minor or absent symptoms, the size of such tumors is often quite large at the time of diagnosis. Therefore, a radical resection of skull base tumors, such as petroclival meningiomas, is often associated with a higher complication rate.

The technical advancements in microsurgery, intraoperative navigation techniques, and addition of electrophysiological monitoring gives the skull base surgeon the possibility of achieving a complete resection by using combined approaches [1,2,3]. However, even experienced skull base surgeons are challenged by these tumors, and surgery is associated with an increased perioperative morbidity [4]. The concept of radical tumor resection has been critically re-assessed, with a focus on minimizing morbidity and preserving the quality of life [5]. In addition, a paradigm change in intracranial high-precision radiation therapy has occurred with the introduction of the Gamma Knife and specialized linear accelerators. This has led to newer therapeutic concepts, and stereotactic radiation is an accepted and reasonable modality in the treatment of partially resected skull base tumors [6,7,8,9].

The concept of planned subtotal resection followed by stereotactic radiotherapy is referred to as adaptive hybrid surgery [10]. The advantage of this concept is, for example, in vestibular schwannoma, to achieve preservation of facial nerve function and hearing. In other benign skull base tumors one can manage inaccessible, recurrent, or residual lesions with radiosurgery [10].

The AHSA (Brainlab^®^, Munich, Germany) software has been developed for planning and performing a surgical resection with adjuvant radiosurgery, to help the neurosurgeon facilitate decision making and, thus, to minimize treatment risks.

We reported our first experience with this innovative software earlier [11]. In our current case series, we analyzed five patients who underwent surgery for benign skull base tumors (vestibular schwannoma, ependymoma, petroclival meningioma, and sphenoid wing meningioma) where, due to size, location, and/or critical adjacent structures, only a partial tumor resection was planned.

## 2. Materials and Methods

Magnetic resonance imaging (MRI) was preoperatively acquired for all patients, and the tumor object and organs at risk (OARs) were defined in the MRI dataset (Figure 1). The tumor object was delineated with the Elements Smartbrush^®^ (Brainlab^®^, Munich, Germany), while the contoured objects for organs at risk (OARs), such as the brainstem, optic nerves, chiasm, optic tract, eyes, and lenses, were obtained by automated atlas segmentation with the Elements^®^ Cranial Segmentation software (Brainlab^®^, Munich, Germany). The automatically generated objects by the software were reviewed for accuracy and edited if needed. An estimated tumor residual was defined by the neurosurgeon using Elements Smartbrush.

The intraoperative neuronavigation system utilized for all patients was Curve™ Image Guided Surgery with the AHSA software (Brainlab^®^, Munich, Germany). The software automatically calculated and displayed a side-by-side comparison of radiosurgery and radiotherapy plans in real-time for the tumor remnants, for three adjuvant treatment strategies:Single fraction stereotactic radiosurgery (sf-SRS);Hypofractionated stereotactic radiosurgery (hf-SRS); andConventional fractionated stereotactic radiotherapy (cf-SRT).

The quality of the treatment plans was thereafter assessed in the software by evaluating the steepness of the dose gradient from tumor coverage to surrounding normal tissue, as specified by a conformity index (CI; Inverse Paddick), and OAR radiation tolerance values, usually evaluated at the mean and maximum values.

On the intraoperative neuronavigation display, the AHSA software provided a summary table inclusive of tumor and OAR parameters. To enable an immediate assessment by the operating neurosurgeon, a color-coded interface is implemented in the software, which follows the principles of a traffic-light pattern and summarizes whether a certain parameter such as tumor coverage or OAR constraints is deemed acceptable, intermediate, or unacceptable (Figure 2): (1) red: desired value is not met, and current value is unacceptable; (2) yellow: desired value is within 10% of acceptability criteria; and (3) green: desired value is met.

Because the toxicity profile for the OARs is evaluated at both mean and maximum dose values, the traffic light shows two concentric disks. The outer disk represents the mean dose value, while the inner disk represents the maximum dose value. Both representations may receive a red, yellow, or green color depending on the dose values that have been defined for the toxicity profile for the OARs. Selection of any structure in the summary table expands the view for this particular organ, to visualize the actual/desired dose in gray.

The AHSA optimization was based on an IMRT algorithm. All three treatment plans were re-normalized post optimization to deliver 100% of the dose to 99% of the volume, with a prescription favoring a CI of 1.1 overdose homogeneity. For sf-SRS, acceptable tumor coverage was defined for a minimum volume of 97% receiving the prescription isodose line. A good conformity was also desired for an acceptable treatment plan and the maximum allowed CI value was 1.4. Cf-SRT required a higher percentage of the volume to receive the prescription isodose line, with a minimum allowed value of 99% of the tumor volume. The maximum allowed CI value for cf-SRT was 1.6. OAR toxicity was defined at the mean and maximum values for most risk organs, with the exception of optical apparatus sub-components, for which an additional dose constraint was specified for 10% of the organ volume.

The intraoperative extent of resection was updated in the AHSA software by using the Elements Intraoperative Structure Update (ISU) software (Brainlab^®^, Munich, Germany). The ISU calculated the residual tumor object by scanning the resection cavity with a calibrated surgical instrument (e.g., a navigation pointer). Continuous ISU was obtained during the resections followed by AHSA. An additional AHSA was also performed for the final residual tumor volume on the postoperative MRI dataset.

## 3. Results

We performed AHSA in five patients presenting with large benign skull base tumors (Table 1). In all patients, a planned partial tumor resection followed by adjuvant SRS of the remnant tumor was planned due to tumor size and nearby critical structures.

Four patients underwent a suboccipital craniotomy for resection of a tumor within the posterior fossa (vestibular schwannoma, ependymoma, petroclival meningioma), and in one patient a temporal approach for resection of a medial sphenoid wing meningioma was performed. The preoperative tumor volumes ranged from 8.52 to 25.2 cm^3^. Intraoperatively, between 3 and 4 ISU were acquired during resection followed by AHSA on the updated remnant tumor volume. The additional surgical time due to intraoperative ISU and AHSA was, on average, 20–30 min. The intraoperative remnant tumor volume measured with the AHSA software was 2.13–12.17 cm^3^ (25–52% of the preoperative tumor volume), with a mean difference of 7–20% in comparison to the effective residual tumor volume measured on the postoperative MR scan. The real extent of resection measured on the postoperative MR scan ranged between 4.79 and 17.72 cm^3^, which corresponded to a 40.7–82.3% resection. The preoperatively estimated residual tumor volume differed in mean 3–10% from the effective residual volume measured on the postoperative MR. The intraoperative automatic AHSA plans of the remaining tumor volume suggested in all five patients that safe radiation would be feasible.

### 3.1. Clinical Outcome

The five cases in our series included skull base tumors with fairly challenging surgical resection. Surgical resection was ended when anatomical circumstances did not allow further resection, and simultaneously AHSA planning confirmed safe hypofractionated or single dose SRS. Table 2 summarizes the preoperative symptoms and outcomes after surgery and radiation. As expected, all patients developed some postoperative neurological deficits, but showed no complications from radiation therapy.

### 3.2. Postoperative Stereotactic Radiosurgery

All our patients who underwent partial resection of a petroclival, sphenoid wing, and cerebellopontine angle meningioma had an intraoperative last ISU with an AHSA suggesting hypofractionated SRS due to the organs’ at-risk constraints. The postoperative radiosurgery plan was performed with iPlan^®^ RT (Brainlab^®^, Munich, Germany), and we compared single fraction (13 Gy) radiosurgery with a hypofractionated scheme (5 × 5 Gy). Finally, treatment consisted of hypofractionated radiotherapy, as suggested by the intraoperative AHSA plan for all three meningiomas.

In another patient, the definite diagnosis revealed an ependymoma (WHO II). In this case, conventional fractionated radiotherapy was recommended by the interdisciplinary tumor board. Due to a stable tumor remnant during the follow-up period, no radiotherapy was necessary. The patient who had a subtotal resection of a vestibular schwannoma experienced subsequent spontaneous regression of the remnant tumor, and therefore preferred to be observed and imaged at regular intervals.

### 3.3. Illustrative Cases

#### 3.3.1. Case 1

A 50-year-old woman presented with aggravated headache, neck pain, and burnout symptoms. Detailed neuropsychological testing confirmed a frontal behavioral syndrome (Figure 3). The patient underwent elective craniotomy and tumor resection with AHSA technology. After complete recovery from the surgery, hypofractionated radiosurgery was performed (5 × 5 Gy) to treat the remnant tumor 1.5 years after resection.

#### 3.3.2. Case 2

A 58-year-old female presented with progressive headache, difficulty swallowing, diplopia, hearing loss, and reduced face sensitivity on the right side (Figure 4). Elective craniotomy and AHSA-assisted tumor resection were performed. The patient underwent hypofractionated radiosurgery (5 × 5 Gy) of the residual meningioma one year after tumor resection.

## 4. Discussion

Despite technical advancements in microneurosurgery, and intraoperative tools with minimally invasive approaches, the challenges and morbidity in skull base surgery remain considerable. During the last decade, quality of life has gained more attention in the treatment of these tumors [12,13]. Due to this continuous paradigm change, partial tumor resection followed by observation or radiation of the residual tumor has become a reasonable concept in the treatment of skull base lesions [14]. Radiosurgery has been proven to be highly effective in long-term tumor control, and is associated with low morbidity in the treatment of meningiomas and other benign skull base tumors, such as vestibular schwannomas [8,15].

Planned subtotal resection in large vestibular schwannomas followed by Gamma Knife radiosurgery has shown excellent preservation of facial nerve and hearing function [16].

The AHSA method was developed with the intention to minimize the morbidity associated with the treatment of complex skull base tumors by integrating surgery and radiosurgery. Herein, we report our first experience with this innovative method in five patients with benign skull base tumors who underwent planned partial tumor resection.

We applied the AHSA software preoperatively to plan and evaluate the intended radiation strategy, by contouring the estimated residual tumor volume. Intraoperatively, several ISU were acquired followed by AHSA on the updated remnant tumor volume. The three obtained different virtual radiosurgery and radiotherapy plans could be compared. Fast, continuous valuable intraoperative feedback of the remnant tumor volume and the resulting radiosurgery plan indicated whether the remaining tumor could be feasibly and safely treated with radiosurgery. The definitive radiosurgery plan correlated quite well with the intraoperatively suggested plan.

Bartek et al. investigated the AHSA software, with respect to dose plans with the Leksell Gamma Plan, and confirmed its usability in vestibular schwannoma [17]. Other authors have shown that the actual and planned postoperative residual tumor volumes were smaller than the ideal radiosurgical target volumes defined by AHSA [18].

Barani et al. demonstrated the feasibility of AHSA for multi-modality management of complex skull base tumors. They concluded that AHSA has the potential to convert cases of conventional fractionated adjuvant radiation to 1–5 fraction radiosurgical cases [19].

Despite its overall accuracy and its capability of reducing postoperative and post-radiation complications, AHSA has some limitations. Firstly, the accuracy of the intraoperative surface scanning of the residual tumor depends on the angle, location, and depth of the remaining tumor, together with the relationship between the pointer and camera visibility. These associated factors may explain why the intraoperative residual tumor volume differed from the effective tumor residual. Additional limitations include: (a) precision of the anatomical mapping can be limited due to intraoperative brain shift, (b) scanning of the remnant tumor is often demanding and depends on angle alignment of the navigation pointer/camera and surgical microscope, (c) intraoperative AHSA volume update is frequently less accurate than intraoperative imaging techniques (for example MRI), and (d) additional intraoperative time is needed to perform AHSA planning.

## 5. Conclusions

Intraoperative SRS evaluation of remnant skull base tumors, applying the AHSA method, is a promising technology. The integration of intraoperative parameters with adjuvant radiation treatment strategies during the resection of skull base tumors might facilitate an optimal multidisciplinary approach and resection strategy, reducing both surgical and radiosurgical risks.

## Figures and Tables

**Figure 1 brainsci-12-01326-f001:**
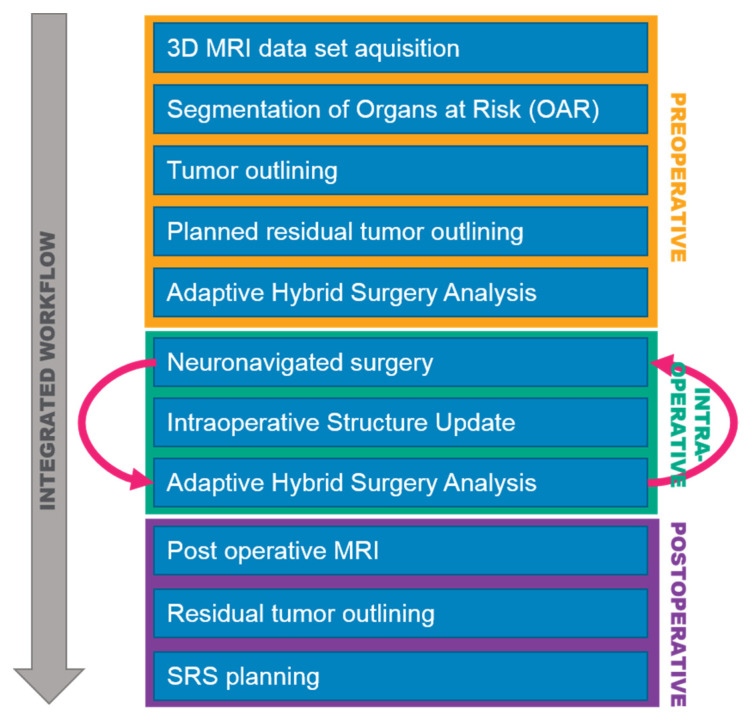
Integrated adaptive hybrid surgery workflow, illustrating the preoperative, intraoperative, and postoperative steps.

**Figure 2 brainsci-12-01326-f002:**
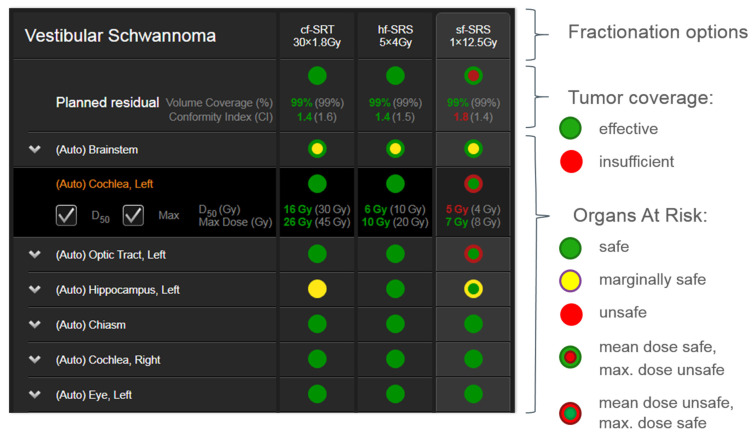
Example of the AHSA summary table depicting the traffic light visualization concept.

**Figure 3 brainsci-12-01326-f003:**
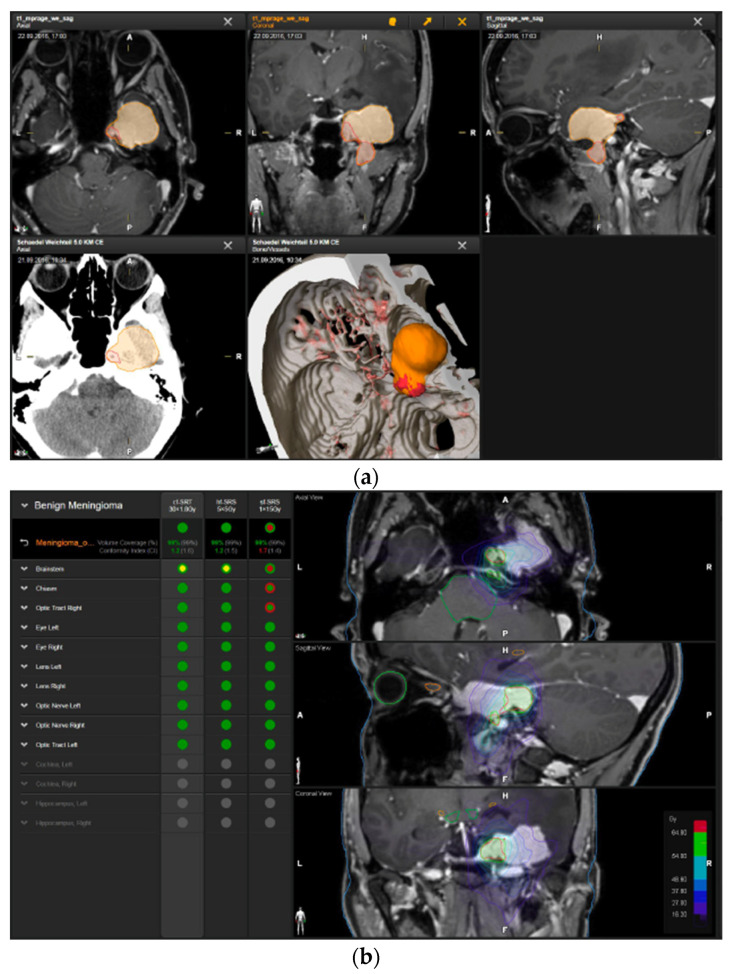
First representative case with AHSA-assisted tumor resection. (**a**) Preoperative MRI imaging of tumor (orange) and planned residual tumor volume (red). (**b**) Dose constraints in the AHSA software for conventional, hypofractionated radiotherapy, and radiosurgery planning for the preoperatively defined residual tumor volume. With this plan, conventional fractionated and hypofractionated radiotherapy were feasible, with effective tumor coverage. 
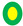
—brainstem: max. dose is marginally safe, whereas mean dose is safe. For single fraction stereotactic radiosurgery, the tumor volume coverage seems effective, while the conformity index was indicating over-treatment. 
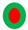
—brainstem: mean dose is safe, while max. dose is unsafe; 
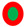
—chiasm: mean dose is unsafe, while max. dose is safe; 
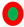
—right optic tract: mean dose is unsafe, while max. dose is safe. (**c**) First intraoperative structure update (ISU) at the time point of 65% residual tumor volume, showing the tumor coverage and organ at risk constraints. At this stage of the resection, AHSA demonstrated that only conventional and hypofractionated radiotherapy were feasible. 
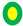
—brainstem: max. dose is marginally safe, whereas mean dose is safe. (**d**) Second ISU acquisition and residual tumor reduction to 58%. (**e**) Third ISU acquisition with a reduction of residual tumor volume to 41%. Dose constraints for conventional, hypofractionated radiotherapy, and radiosurgery are demonstrated. At the final stage of the resection, the conventional and hypofractionated radiotherapy organ risk constraints were unchanged. 
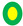
—brainstem shows that max. dose is marginally safe, whereas mean dose is safe. The single-dose stereotactic radiosurgery constraints show that: 
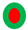
—brainstem: mean dose safe, max. dose unsafe; 
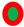
—chiasm: mean dose unsafe, max. dose safe; 
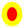
—right optic tract: mean dose marginally safe, max. dose unsafe. (**f**) Fusion of intraoperative CT to final intraoperative ISU. (**g**) Preoperative and 3 months postoperative MRI imaging for planning of radiosurgery (5 × 5 Gy).

**Figure 4 brainsci-12-01326-f004:**
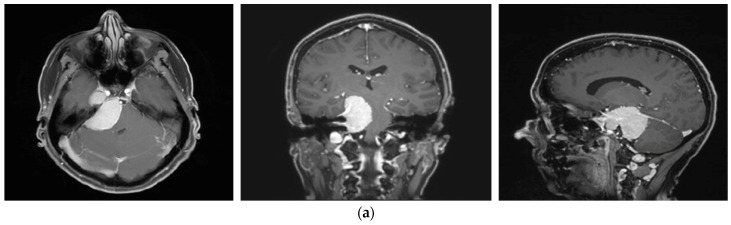
Second representative case of AHSA-supported tumor resection. (**a**) MRI imaging of petroclival meningioma (axial, coronal, sagittal). (**b**) Depiction of preoperative tumor volume (orange) and planned residual tumor volume (red). (**c**) AHSA summary table showing the stereotactic radiation constraints for the preoperatively planned residual tumor volume. With this plan, conventional fractionation and hypofractionation were feasible with effective tumor coverage. 
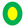
—brainstem: mean dose is safe, while the max. dose is marginally safe; 
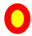
—right optic tract: mean dose unsafe, max. dose is marginally safe; 
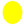

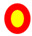
—right hippocampus: marginally safe, and mean dose unsafe, max. dose is marginally safe. (**d**) First intraoperative structure update (ISU) with a residual tumor volume of 82% with the calculated dose constraints for conventional and hypofractionated radiotherapy. Single dose stereotactic radiosurgery was not feasible with this degree of remaining tumor. At this point, the dose constraints for conventional radiotherapy were: 
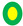
—brainstem: mean dose is safe, max. dose is marginally safe; 
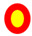
—right hippocampus: mean dose unsafe, max. dose is marginally safe. Hypofractionated radiotherapy: 
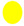
—brainstem: marginally safe; 
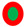
—right cochlea: mean dose unsafe, max. dose safe; 
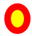
—right optic tract: mean dose unsafe, max. dose is marginally safe; 
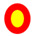
—right hippocampus: mean dose unsafe, max. dose is marginally safe. (**e**) Second intraoperative ISU with residual tumor volume of 74% and calculated dose constraints for hypofractionated radiotherapy. Single dose stereotactic radiosurgery was still not considered feasible with this residual tumor volume. The dose constraints for organs at risk for conventional and hypofractionated radiotherapy were unchanged compared to the first ISU. (**f**) Third intraoperative ISU with residual tumor volume of 47% and calculated dose constraints for conventional, hypofractionated radiotherapy, and radiosurgery. The current dose constraints for organs at risk were the following for conventional radiation: 
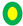
—brainstem: mean dose is safe, max. dose is marginally safe; 
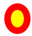
—right hippocampus: mean dose unsafe, max. dose is marginally safe. Hypofractionated radiation, which was unchanged for the first and second ISU: 
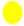
—brainstem: marginally safe; 
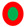
—right cochlea: mean dose unsafe, max. dose safe; 
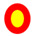
—right optic tract: mean dose unsafe, max. dose is marginally safe; 
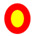
—right hippocampus: mean dose unsafe, max. dose is marginally safe. For single fraction radiosurgery, the OAR dose constraints were available but considered to be unsafe. 
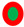
—brainstem: mean dose unsafe, max. dose safe; 
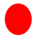
—chiasma: unsafe; 
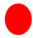
—right cochlea: unsafe; 
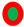
—left optic tract: mean dose unsafe, max. dose safe; 
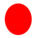
—right optic tract: unsafe; 
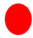
—right hippocampus: unsafe. (**g**) Overlay of preoperatively estimated and intraoperative effective residual tumor volume in AHSA. (**h**) Final intraoperative dose constraints after last ISU and data fusion with intraoperative CT. The dose constraints for OARs appeared to improve and were as follows for conventional and hypofractionated radiotherapy: 
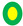
—brainstem: mean dose is safe, max. dose is marginally safe. 
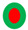
—brainstem: mean dose safe, max. dose unsafe; 
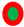
—chiasm: mean dose unsafe, max. dose safe; 
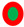
—right optic tract: mean dose unsafe, max. dose safe. (**i**) Comparison of pre- and 3 months postoperative MRI for stereotactic radiation planning. The residual tumor was finally treated with hypofractionated radiotherapy (5 × 5 Gy).

**Table 1 brainsci-12-01326-t001:** MRI-based values of pre- and postoperative tumor volumes, including extent of resection. EOR = Extend of resection.

Case No.	Diagnosis	Preop. TuVol (cm^3^)	Estim Resid TuVol (cm^3^)	AHSA Remnant TuVol (cm^3^)	Postop TuVol (MR) cm^3^	EOR (Postop. MR) (cm^3^)	EOR (%)
1	Sphenoid wing meningioma WHO I	22.54	5.00	8.89	4.82	17.72	78.6
2	Petroclival meningioma WHO I	25.5	10.63	12.17	7.98	17.52	68.7
3	Cerebellopontine angle ependymoma WHO II	20.8	8.05	6.46	8.56	12.24	58.8
4	Vestibular schwannoma KOOS IV WHO I	8.52	1.56	2.13	1.51	7.01	82.3
5	Cerebellopontine angle meningioma WHO I	11.8	6.49	6.19	7.01	4.79	40.7

**Table 2 brainsci-12-01326-t002:** Initial symptoms at presentation, new neurological deficits after surgery, and complications of radiation therapy after the indicated follow-up time.

Diagnosis	Initial Symptoms	New Neurological Deficits after Tumor Resection	Postoperative Development of Preoperative Symptoms	Postoperative Follow Up Time	Complications from Stereotactic or Conventional Radiation Therapy
Sphenoid wing meningioma	Aggravating head and neck pain, fatigue, diagnosis of burnout, fronto-limbic cognitive dysfunction	Postoperative progressive edema and swelling with uncal herniation, requiring emergency craniectomy and prolonged recovery Homonymous superior hemianopsia Left facial nerve palsy House Brackmann II Right hyposensitivity V1–V3	Reduced frequency of head and neck pain	72 months	none
Petroclival meningioma	Progressive head pain, progressive swallowing difficulties, hypesthesia right side, double vision, hypesthesia V1–V3 right side, mild left hemiparesis, sleep disorder, impaired concentration	Right complete deafness and failure of vestibulocochlear nerve Right facial nerve palsy House Brackmann III	Reduced frequency of head pain, no more swallowing difficulties, complete remission of hemiparesis	72 months	none
Cerebellopontine angle ependymoma	Gait instability, progressive right hypacusis, vertigo	Right complete deafness, right facial nerve palsy, House Brackmann II	Unchanged	68 months	N/A
Vestibular schwannoma	Progressive gait instability, progressive left hypacusis, tinnitus, left hypesthesia V1–V2, facial palsy House Brackmann II	Progressive facial palsy House Brackmann III, slight swallowing issues	Improvement of gait ability, complete remission of tinnitus	70 months	N/A
Cerebellopontine angle meningioma	Incidental finding with brain stem compression	Slight left hypoacusis	Unchanged	66 months	none

## Data Availability

The data presented in this study are available on request from the corresponding author.

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
