# Peer review of "Adaptive Hybrid Surgery Experiences in Benign Skull Base Tumors"

_brainsci, 2022, doi:10.3390/brainsci12101326_

Round 1
Reviewer 1 Report
The authors of this manuscript describe their use in five patients of an innovative intraoperative technique that assesses the amount of a skull-base tumor that needs to be removed so that some type of radiation therapy (radiosurgery, fractionated radiation therapy) can be used with low risk to surrounding neural and vascular structures. The technique is nicely described but I have several comments and concerns.
1. The manuscript would benefit from a table (and text) indicating how long the five patients were followed and whether or not they experienced any complications from either surgery or radiation therapy.
2. Page 2, line 53. The sentence is confusing. I would change it to: "In addition, a paradigm change in high-precision radiation therapy has occurred with the introduction of...." and place it within the paragraph above.
3. Page 2, lines 52-55.Please change the word "lead" to "led" and place a comma after the word "concepts".
4. Page 2, line 57. I believe the abbreviation "STS" should be "SRS" and, in any event, should be spelled out "stereotactic radiosurgery".
5. Page 2, line 70. This is the first time magnetic resonance imaging has been mentioned. Please change sentence to begin: "Magnetic resonance imaging (MRI)..."
6. Page 2, line 76. What do you mean by stating that "the automatically generated objects were controlled"? Please clarify.
7. Page 2, line 85. Please change the word "conventionally" to "conventional"
8. Page 3, line 98. Please change the word "Since" to "Because"
9. Page 3, line 126. Please add a comma after the parenthesis.
10. Page 3, line 129. Please add a comma after the word "patient". Also, please add a comma after the word "case" and delete the word "a" so that the sentences reads "...another patient, the definite diagnosis revealed an ependymoma WHO II. In this case, conventional..."
11. Page 3, lines 132-133. Please change the term "revealed a" to "experienced", change the word "consecutive" to "subsequent", and change the phrase "a wait and scan strategy" to "to be observed and imaged at regular intervals" so that the sentence reads: "...of a vestibular schwannoma experienced subsequent spontaneous regression of the remnant tumor and therefore preferred to be observed and imaged at regular intervals"
12. Results section. Here is where some type of baseline and follow-up information is needed, ideally in both text and a table, including initial presenting signs, any complications of surgery, any improvement or worsening of signs after surgery, how long were the five patients followed? Were there any complications from radiation therapy (after all, that's the whole point of the technique!)
13. Figure 3G. The three MR images have been reversed. Please correct.
14. Figures that show the dose constraints (3b,c an e; 4c,d,e,f, and h) would benefit from legends that emphasize what is being shown; ie, what structures/organs are subject to what severity of damage at this stage of the procedure.
15. Page 12, line 230. Please change the word "preserving" to "preservation"
16. Page 12, line 233. As begun, this sentence seems to refer to an earlier publication whereas it relates to this series of patients. I would change the beginning to: "Herein, we report our first experience with...."
17. Page 12, line 243. Please delete the words "that had" and change the phrase "in comparison to" to "with respect to" so that the sentence reads: "Bartek et al. investigated the AHSA software with respect to dose plans..."
18. Page 12, line 245. Please delete the comma after the word "shown"
19. Page 12, line 248. Please change the word "the" to "they" and delete the comma after the word "concluded".
20. Page 12, line 249. Please change the phrase "conventionally-fractionated" to conventional fractionated"
21. Page 12, lines 251-255. You are now pointing out the limitations of this technique. There needs to be acknowledgement of this issue. Instead of beginning the sentence with the word "However", I suggest some type of introductory sentence; eg, "Despite it's overall accuracy and its capability of reducing postoperative and post-radiation complications, AHSA has some limitations. Firstly, the accuracy....visibility. These associated factors can explain why [no comma after the word explain] the intraoperative residual tumor volume differed from the effective tumor residua. Other limitations include: a) prescision..."
Author Response
Response to reviewers
We would like to thank the reviewer for his/her helpful comments and suggestions, that helped us to improve the quality of our manuscript.
Title of the Manuscript: Adaptive hybrid surgery experience in benign skull base tumors
Manuscript Number: brainsci-1907626
Reviewer 1
Comment 1:
The manuscript would benefit from a table (and text) indicating how long the five patients were followed and whether or not they experienced any complications from either surgery or radiation therapy.
Authors’ Response We thank the reviewer for this important suggestion. We added the requested information in table 2 and the results section.
Comment 2:
Page 2, line 53. The sentence is confusing. I would change it to: "In addition, a paradigm change in high-precision radiation therapy has occurred with the introduction of...." and place it within the paragraph above.
Authors’ Response: Thank you for this suggestion, the sentence was changed.
Comment 3:
Page 2, lines 52-55. Please change the word "lead" to "led" and place a comma after the word "concepts".
Authors’ Response: We adjusted the text accordingly.
Comment 4:
Page 2, line 57. I believe the abbreviation "STS" should be "SRS" and, in any event, should be spelled out "stereotactic radiosurgery".
Authors’ Response: Thank you for this comment, we made the corrections.
Comment 5:
Page 2, line 70. This is the first time magnetic resonance imaging has been mentioned. Please change sentence to begin: "Magnetic resonance imaging (MRI)..."
Authors’ Response: We edited the sentence as proposed by the Reviewer.
Comment 6:
Page 2, line 53. The sentence is confusing. I would change it to: "In addition, a paradigm change in high-precision radiation therapy has occurred with the introduction of...." and place it within the paragraph above.
Authors’ Response: Thank you for this suggestion, the change has been done.
Comment 7:
Page 2, line 76. What do you mean by stating that "the automatically generated objects were controlled"? Please clarify.
Authors’ Response: The automatically generated objects by the software were reviewed for accuracy and edited if needed.
Comment 8:
Page 2, line 85. Please change the word "conventionally" to "conventional"
Authors’ Response: Thank you for this suggestion, this change was made.
Comment 9:
Page 3, line 98. Please change the word "Since" to "Because"
Authors’ Response: The correction was done.
Comment 10:
Page 3, line 126. Please add a comma after the parenthesis.
Authors’ Response: Thank you for this comment, the comma was added.
Comment 11:
Page 3, line 129. Please add a comma after the word "patient". Also, please add a comma after the word "case" and delete the word "a" so that the sentences reads "...another patient, the definite diagnosis revealed an ependymoma WHO II. In this case, conventional..."
Authors’ Response: Thank you for this helpful advice, the sentence was edited accordingly.
Comment 12:
Page 3, lines 132-133. Please change the term "revealed a" to "experienced", change the word "consecutive" to "subsequent", and change the phrase "a wait and scan strategy" to "to be observed and imaged at regular intervals" so that the sentence reads: "...of a vestibular schwannoma experienced subsequent spontaneous regression of the remnant tumor and therefore preferred to be observed and imaged at regular intervals"
Authors’ Response: Thank you for this useful comment, the sentence was adjusted as you proposed.
Comment 13:
Results section. Here is where some type of baseline and follow-up information is needed, ideally in both text and a table, including initial presenting signs, any complications of surgery, any improvement or worsening of signs after surgery, how long were the five patients followed? Were there any complications from radiation therapy (after all, that's the whole point of the technique!)
Authors’ Response: Thank you for this comment. We added a new table with clinical information and also added this to the results section.
Comment 14:
Figure 3G. The three MR images have been reversed. Please correct.
Authors’ Response: We updated the figure 3G.
Comment 15:
Figures that show the dose constraints (3b,c an e; 4c,d,e,f, and h) would benefit from legends that emphasize what is being shown; ie, what structures/organs are subject to what severity of damage at this stage of the procedure.
Authors’ Response: Thank you for this suggestion, we adjusted the figures and added the information of dose constraint for the OARs.
Comment 16:
Page 12, line 230. Please change the word "preserving" to "preservation"
Authors’ Response: Thank you for this suggestion, we exchanged the word.
Comment 17:
Page 12, line 233. As begun, this sentence seems to refer to an earlier publication whereas it relates to this series of patients. I would change the beginning to: "Herein, we report our first experience with...."
Authors’ Response: Thank you for this suggestion, the change has been done.
Comment 18:
Page 12, line 233. As begun, this sentence seems to refer to an earlier publication whereas it relates to this series of patients. I would change the beginning to: "Herein, we report our first experience with...."
Authors’ Response: Thank you for this comment, the change has been done.
Comment 19:
Page 12, line 243. Please delete the words "that had" and change the phrase "in comparison to" to "with respect to" so that the sentence reads: "Bartek et al. investigated the AHSA software with respect to dose plans..."Authors’ Response: Thank you for this suggestion, the change has been done.
Comment 20:
Page 12, line 245. Please delete the comma after the word "shown"
Authors’ Response: Thank you for this suggestion, we deleted the comma.
Comment 21:
Page 12, line 248. Please change the word "the" to "they" and delete the comma after the word "concluded".
Authors’ Response: Thank you for this suggestion, the edits have been fulfilled.
Comment 22:
Page 12, line 249. Please change the phrase "conventionally-fractionated" to conventional fractionated"
Authors’ Response: Thank you for this suggestion, the phrase has been changed accordingly.
Comment 23:
Page 12, lines 251-255. You are now pointing out the limitations of this technique. There needs to be acknowledgement of this issue. Instead of beginning the sentence with the word "However", I suggest some type of introductory sentence; eg, "Despite it's overall accuracy and its capability of reducing postoperative and post-radiation complications, AHSA has some limitations. Firstly, the accuracy....visibility. These associated factors can explain why [no comma after the word explain] the intraoperative residual tumor volume differed from the effective tumor residua. Other limitations include: a) prescision..."
Authors’ Response: We would like to thank the reviewer for this final comment. We changed the section on limitations according to the suggestion.
Reviewer 2 Report
Manuscript: "Adaptive hybrid surgery experience in benign skull base tumors: case series and technical note", submitted by Jenny Christine Kienzler and Javier Fandino, focuses on multimodality treatment approaches that include controlled partial tumor resection followed by radiosurgery. This approach protects the brain from a serious threat to the patient's health and life.
The article is not well structured and contains many unclear sentences. The introduction, results, and discussion are not well written, the Figures are in the wrong format, sparsely described, and difficult to analyze. Grammatical errors and ambiguous descriptions are also found in several places. The section on materials and methods should be accurately described, as required by the journal. The work requires thorough proofreading. It is difficult to read and lacks many important elements as required for a scientific paper.
Author Response
Reviewer 2
Comment 1: The article is not well structured and contains many unclear sentences. The introduction, results, and discussion are not well written, the Figures are in the wrong format, sparsely described, and difficult to analyze. Grammatical errors and ambiguous descriptions are also found in several places. The section on materials and methods should be accurately described, as required by the journal. The work requires thorough proofreading. It is difficult to read and lacks many important elements as required for a scientific paper.
Authors’ Response: We thank the reviewer for this comment. In the meantime, the manuscript has undergone editing and proofreading. We hope it will fulfill the reviewer's expectations.
Reviewer 3 Report
In this paper, the authors reported a small case series of five complex skull base cases in which AHSA (Brainlab®, Munich, Germany) software was used. Overall, the concept of planned subtotal resection followed by STS with the aim to preserve the quality of life and reduce the risk of postoperative neurological impairment is nowadays the main trend in complex skull base procedures. The application of software to plan and help the neurosurgeon in decision making and thus minimize treatment risks is interesting. However, in my opinion, there are some issues:
1. The Abstract section needs to be more concise. It should be rewritten focusing on the main findings and results of the paper.
2. It is not clear how AHSA helped during the resection: do the authors decided to not proceed with surgical removal based on planned postoperative SRS? Have there never been any technical reasons why we had to stop earlier?
3. What were the clinical outcomes of the selected cases?
4. The authors presented a very small cohort, why publish only a case series in which no statistical data about the clinical-radiological outcomes are provided?
Author Response
Reviewer 3
Comment 1: The Abstract section needs to be more concise. It should be rewritten focusing on the main findings and results of the paper.
Authors’ Response: We would like to thank the reviewer for this comment. The abstract was edited accordingly.
Comment 2: It is not clear how AHSA helped during the resection: do the authors decided to not proceed with surgical removal based on planned postoperative SRS? Have there never been any technical reasons why we had to stop earlier?
Authors’ Response: Thank you for this important question. We added a paragraph on this topic (3.1). Surgical resection was in our series stopped, when anatomical circumstances did not allow for further resection, and AHSA planning confirmed safe hypofractionated treatment or even single dose radiosurgery. We did not stop the surgery due to technical reasons.
Comment 3: What were the clinical outcomes of the selected cases?
Authors’ Response: We would like to thank the reviewer for this comment. We added a section (3.1 Clinical outcome) and a table (table 2) describing the clinical outcome.
Comment 4: The authors presented a very small cohort, why publish only a case series in which no statistical data about the clinical-radiological outcomes are provided?
Authors’ Response: We thank the reviewer for this important point. We present a case series of rare and eloquent skull base tumors. These tumors are rather rare. The use of the AHSA software is only indicated in fairly demanding cases, where a complete resection is not feasible from the beginning of planning the surgery. Also, the addition of more case would not change our conclusion and second, a lot of case would have to be added to get reliable statistic results.
Round 2
Reviewer 1 Report
The authors have satisfactorally addressed my comments and recommendations.
Reviewer 2 Report
Thank you for correcting and completing the draft. I recommend acceptance of this manuscript for publication.
Reviewer 3 Report
I think that the manuscript is now suitable for publication